# Systemic and CSF Interleukin-1α Expression in a Rabbit Closed Cranium Subarachnoid Hemorrhage Model: An Exploratory Study

**DOI:** 10.3390/brainsci9100249

**Published:** 2019-09-24

**Authors:** Davide Marco Croci, Stefan Wanderer, Fabio Strange, Basil E. Grüter, Daniela Casoni, Sivani Sivanrupan, Hans Rudolf Widmer, Stefano Di Santo, Javier Fandino, Luigi Mariani, Serge Marbacher

**Affiliations:** 1Department of Neurosurgery, University Hospital Basel, 4031 Basel, Switzerland; luigi.mariani@usb.ch; 2Cerebrovascular Research Group, Department of BioMedical Research, University of Bern, 3008 Bern, Switzerland; stefan.wanderer@ksa.ch (S.W.); Fabio.strange@ksa.ch (F.S.); Sivani.sivanrupan@unibe.ch (S.S.); javier.fandino@ksa.ch (J.F.); Serge.Marbacher@ksa.ch (S.M.); 3Department of Neurosurgery, Neurocenter of Southern Switzerland, Regional Hospital Lugano, 6900 Lugano, Switzerland; 4Department of Neurosurgery, Kantonsspital Aarau, 5001 Aarau, Switzerland; 5Department of Biomedical Research, University of Bern, 3008 Bern, Switzerland; Daniela.Casoni@unibern.ch; 6Department of Neurosurgery, Bern University Hospital, Inselspital Bern, 3008 Bern, Switzerland; Hansrudolf.widmer@insel.ch (H.R.W.); Stefano.disanto@insel.ch (S.D.S.)

**Keywords:** subarachnoid hemorrhage, IL-1α, inflammation, animal model

## Abstract

Background: The inflammatory pathway in cerebrospinal fluid (CSF) leads to delayed cerebral vasospasm (DCVS) and delayed cerebral ischemia (DCI) after subarachnoid hemorrhage (SAH). The role of IL-1α has never been evaluated in a rabbit SAH model. The aim of our study is to analyze systemic and CSF changes of IL-1α, and to evaluate potential associations with the onset of DCVS in a rabbit closed cranium SAH model. Methods: 17 New Zealand white rabbits were randomized into two groups, SAH (*n* = 12) and sham (*n* = 5). In the first group, SAH was induced by extracranial-intracranial shunting from the subclavian artery into the cerebral cistern of magna under intracranial pressure (ICP) monitoring. The sham group served as a control. The CSF and blood samples for IL-1α measurement were taken at day zero before SAH induction and at day three. Results: There was a significant increase of ICP (*p* = 0.00009) and a decrease of cerebral perfusion pressure (CPP) (*p* = 0.00089) during SAH induction. At follow up, there was a significant increase of systemic IL-1α in the SAH as compared with the sham group (*p* = 0.042). There was no statistically significant difference in the CSF values in both groups. The CSF IL-1α values showed a correlation trend of DCVS. Conclusions: Systemic IL-1α levels are elevated after SAH induction in a rabbit SAH model.

## 1. Introduction

Delayed cerebral vasospasm (DCVS) and delayed cerebral ischemia (DCI) are severe complications of subarachnoid hemorrhage (SAH). The inflammatory pathway due to blood hemolysis in cerebrospinal fluid (CSF) has been recognized to be one of the leading factors causing DCVS and DCI. Different cytokines and interleukins have been described as mediators of the inflammation cascade initiated by blood products in the subarachnoid space, leading to DCVS and DCI [1,2,3,4]. Interleukin-1 (IL-1) is a family of cytokines which induces a group of different cytokines and the expression of integrins on leukocytes and endothelial cells mediating the inflammatory response [5,6,7]. IL-1 is produced in different immune cells such as macrophages, lymphocytes, and microglia [8,9]. Despite IL-1 being typically linked to inflammation, it has also been associated with different functions such as insulin secretion, fever induction, and neuronal phenotype development [10,11,12,13,14]. IL-1 has been described to be a key mediator of neuronal injury after acute brain injury [15,16]. Moreover, it usually upregulates the expression of interleukin-6 (IL-6), which triggers local inflammation and activation of the systemic acute phase response. IL-1 stimulates the release of IL-6 from mast cells [17], which have been described to be present in the aneurysm wall of SAH patients and the muscular layer of cerebral arteries [18,19]. The IL-6 concentrations are elevated in the cerebrospinal fluid (CSF) in animal models of SAH and in patients suffering SAH. Moreover, high concentrations of CSF IL-6 correspond to worse clinical outcomes in patients affected by SAH [1,3].

IL-1α and IL-1β are the most known interleukins subtypes in the IL-1 family and both are proinflammatory and bind at the same receptor, the type I IL-1 receptor (IL-1RI). The initiated signaling cascade results in the expression of inflammatory genes [5,6,20]. IL-1α differs mainly from IL-1β in primary structure. IL-1α, in contrast to IL-1β, appears to have bioactivity while in the form of an intact pro-cytokine precursor on the cell surface and intracellularly [21]. The IL-1 receptor antagonist (IL-1Ra) blocks the signaling at the receptor, inhibiting the inflammatory effects of IL-1α and IL-1β [22,23]. IL-1α has an emerging importance in the initiation and maintenance of inflammation in different human diseases and the initiation of the sterile inflammatory response. In ischemic strokes, IL-1α has been described to be expressed early in areas of focal neuronal injury after ischemic injury. In addition, it is chronically elevated in the brain after an experimental stroke, suggesting that it is present during post-stroke angiogenic periods. Moreover, IL-1α not only precedes the expression of IL-1β and IL-6, but it has also been described to be more potent in stimulating the expression of IL-6 [24,25,26].

Considering that to date no medical treatment has been shown to properly prevent the onset of DCVS and DCI after SAH, the outcome of patients affected by DCVS and DCI after SAH remains very poor. As previously described, the inflammation reaction initiated in early brain injury phase after SAH is highly involved in the development of DCVS and DCI [1,3,18,27]. IL-1α is one of the key player and the earliest interleukins released during the sterile inflammation cascade [24]. A better understanding of this interleukin in the inflammation cascade after SAH could be essential for the development of future target therapies to prevent DCVS and DCI in patients affected by SAH.

With this study, we aimed to evaluate the levels of systemic and CSF IL-1α in an established extracranial intracranial closed rabbit SAH model and to analyze if there are any correlations with DCVS.

## 2. Materials and Methods

### 2.1. Animals, Study Design, and Anesthesia

As a subproject of an ongoing study, a total of 17 female New Zealand white rabbits (3.1–4.1 kg body weight, 16 weeks old, Charles River, Sulzfeld, Germany) were previously randomly allocated into two different groups using a web-based randomization system [28], i.e., the SAH group (*n* = 12) and in the sham group (*n* = 5). After the clinical examination, general anesthesia was induced in all rabbits with a mixture of subcutaneous ketamine (Narketan 100 mg/mL, Vetoquinol AG, Bern, Switzerland) (30 mg/kg) and xylazine (6 mg/kg, Xylapan, 20 mg/mL, Vetoquinol AG, Bern, Switzerland). After 15 min under administration of 2–4 L of O_2_ through a facial mask, two 22 G intravenous cannulae were inserted in the marginal auricular vein and into the auricular artery, respectively. Perincisional ropivacaine 1% (Sintetica S.A., Mendrisio, Switzerland) was infiltrated into the axillary region (access for the subclavian artery). A supraglottic device was introduced and general anesthesia maintained with isoflurane in oxygen targeting an EtIso of 1.3%. Spontaneous ventilation was allowed. Ringer lactate (4–10 mL/kg/h) was infused through the vein access. Continuous monitoring of heart rate, respiratory rate, oxygen arterial saturation, capnography, invasive blood pressure, non-invasive blood pressure (Doppler technique), esophageal temperature, as well as inspired and expired fraction of gases (air, CO_2_ and isoflurane) was provided. During the procedure, at least 1 blood gas was analyzed.

Additional boluses of fentanyl (Fentanyl Syntetica 0.5 mg/10 mL, Sintetica S.A., Mendrisio, Switzerland) were provided (3–10 mcg/kg IV) if nociception was deemed insufficient. Nociception was continuously assessed through cardiovascular monitoring and at intervals of 5 min through stimulation of pedal reflex (toe pinch). Hypothermia was prevented using a warming mattress and hypotension (MAP less than 60 mmHg) was addressed with the use of noradrenaline titrated to effect. Postoperative analgesia was provided with a fentanyl patch (12 mcg/h; Durogesic Matrix 12 µg/h, Janssen-Cilag AG, Schaffausen, Switzerland) put on the outer ears. The animals were carefully observed during the recovery phase. Oxygen supplementation and fluid therapy were provided until sternal recumbency was achieved. Neurological status was assessed at 6, 12, 24, 48, and 72 h post SAH, according to a four-point grading system, as previously described [29].

### 2.2. Digital Subtraction Angiography and SAH Induction

Detailed protocols for surgery and angiography have been previously described elsewhere [30,31,32]. Digital subtraction angiography (DSA) was performed under general anesthesia at day 0 prior to SAH and on day 3. The rabbits’ subclavian artery was prepared and cannulated with a 5.5 French catheter (Silicone Catheter STH-C040, Connectors Verbindungstechink AG, Tagelswangen, Switzerland). The catheter tip was advanced to the origin of the vertebral artery. Intraarterial bolus injection of contrast dye followed (0.6 mL/kg Iopamidol, Iopamiro, Bracoo Suisse, Mendrisio, Switzerland). Images of the basilar artery were obtained using a rapid sequential angiographic technique (DFP 2000A, Toshiba, Tokyo, Japan). The same procedure was performed at day 3 for follow up. The caliber of the basilar artery (BA) was measured, as previously described (the midpoint of the basilar artery, and 0.5 cm above and below), in a blinded manner using ImageJ (National Institutes of Health, Bethesda, MD, USA) [1,33]. To assess the degree of DCVS, the relative change between baseline and follow up was compared [1,30,34]. Following baseline DSA on day 0, either induction of SAH or sham procedure was performed, as described earlier [30,31,32,35,36]. In all animals, intracranial pressure (ICP), arterial blood pressure, and respiration rate were continuously monitored. An ICP probe (Codman Disposable ICP Kit, Spreitenbach, Switzerland) was inserted through a right frontal osteotomy [37]. Arterial blood gas analysis was performed prior to angiography (ABL 725, Radiometer, Copenhagen, Denmark). In prone position, a 27 G spinal access needle was inserted into the cisterna magna. In animals assigned to the SAH, the needle was connected to the catheterized subclavian artery to induce a hemorrhage [30,31]. Sham-operated animals underwent puncture of the cisterna magna, CSF sampling (1.5 mL), and CSF replacement with 1.5 mL artificial CSF (ACSF, Tocris Bioscience, Bristol, UK).

### 2.3. Blood and CSF Samples Analysis

CSF and blood samples were taken at day 0 and day 3. At day 0, 1.5 mL CSF was aspirated after puncture. The blood samples were collected together with the blood gas analysis using EDTA-coated tubes. The samples were centrifuged with 1500× *g* for 15 min at 4 °C. The supernatant was segregated and stored at −80 °C until measurement. For IL-1α quantification, a specific IL-1α enzyme-linked immunosorbent assay (ELISA) kit was used (Cayman Chemical, 1180 E. Ellsworth Rd, Ann Arbor, MI, USA). ELISA was performed according to the manufacturers’ protocol. Euthanasia was performed on day 3 by injection of 40 mg/kg sodium thiopental (Pentothal, Ospedalia, Hünenberg, Switzerland).

### 2.4. Statistical Analysis

Data were analyzed and visualized using IBM SPSS statistical software version 21.0 (IBM Corp., New York, NY, USA). Continuous values were given as mean ± SD, if not otherwise indicated. The comparison of baseline and follow up was compared by the Wilcoxson signed rank test (nonparametric, repeated measures). The differences between the normally distributed IL-1α measures and baseline values of two groups and the neurological scores were analyzed by Student’s *t*-test. A *p*-value < 0.05 was regarded as statistically significant. Correlations were calculated by the Pearson’s correlation test.

### 2.5. Ethic Approval

The study was performed in accordance with the local guidelines for the care and use of experimental animals. The project was performed according to the Animal Research: Reporting of In Vivo Experiments (ARRIVE) guidelines [38] and was performed in accordance with the National Institutes of Health Guidelines for the care and use of experimental animals and with the approval of the Animal Care Committee of the Canton Bern, Switzerland (Approval Nr. BE58/17). A power analysis was not applicable because of the exploratory nature of the study and because it was part of a subproject of a larger study.

## 3. Results

### 3.1. Physiological Parameters, ICP Time Course, and SAH and Clinical Scores

A total of 14 (10 in the SAH and four in the sham group) out of 17 rabbits reached the primary endpoint. One rabbit was prematurely excluded after it died before the start of the group-specific procedures. In two rabbits, premature euthanasia was performed due to a severe postoperative neurological deficit after SAH induction. Arterial blood gas analysis was invalid for one animal in each of the SAH and sham operated group during follow up. There were no complications either related to wound healing, cerebrospinal fluid leakage, or infections along the frontal osteotomy sites, subclavian skin incision, or nuchal cisterna injection point.

Arterial blood gas values and basic physiological parameters were within a normal range and there was no significant difference between the groups at baseline and follow up, except for potassium levels which were significantly higher at follow up in the sham operated group (Table 1).

All animals in the SAH group showed a significantly marked ICP increase during SAH induction from baseline to peak (*p* = 0.00009), with a corresponding decrease of cerebral perfusion pressure (CPP) (*p* = 0.00089) (Figure 1, panel left). During SAH induction, there were no significant changes of middle arterial pressure (MAP) and respiratory rate (RR) between the baseline and the peak. There was a trend of worse neurological outcomes during the postsurgical period in the SAH group without significant differences, *p =* 0.27 *(*Figure 1, panel right).

All surviving rabbits in the SAH group demonstrated extensive coagulated diffuse subarachnoid blood which resulted in moderate grades of SAH with a mean bleeding sum score [34] of 5.9 (±2.1) in the SAH group as compared to none in the sham group (*p* = 0.00002).

### 3.2. Angiographic Delayed Cerebral Vasospasm

At day zero, the baseline angiography in the SAH group showed a mean BA diameter of 310 (±50) µm. At follow up, measurements showed a significantly decreased diameter with a mean of 220 (±60) µm, *p* = 0.0001. The sham group showed a mean BA diameter of 410 (±60) µm at baseline and 450 (±70) µm at follow up (*p* = 0.446), Figure 2.

### 3.3. CSF and Systemic IL-1α Levels

Notably, overall CSF IL-1α values were significantly higher than serum IL-1α values (*p* < 0.0001). Therefore, we measured CSF IL-1α levels in the range of 5–25 pcg/mL in the CSF while levels in serum were below 1 pcg/mL (Figure 3). In the SAH group, there was a trend of higher CSF IL-1α levels at baseline and follow up as compared with the sham group without statistical significance (Figure 3). In the serum, there was a trend of increasing IL-1α levels between baseline and follow up in the SAH group. When comparing the follow-up values in the sham and SAH groups there was a significant increase of systemic IL-1α in the SAH group as compared with the sham group (*p* = 0.042). The CSF-IL-1α values showed a trend of a negative correlation (Pearson’s *r* = −0.25, *p* = 0.410) with the angiographic diameter of the basilar artery. No significant correlation between serum IL-1α values and DCVS could be found (Pearson’s *r* = −0.03, *p* = 0.856).

## 4. Discussion

The results of this study demonstrated that SAH was well induced with a significant increase of ICP and a decrease of CPP. DCVS was significantly present in the SAH group. The CSF IL-1α values were overall significantly higher than serum IL-1α values. Moreover, at follow up, the SAH group showed a significantly higher level of IL-1α in serum. A correlation trend between CSF IL-1α and DCVS was found.

Currently, there are neither prophylactic nor therapeutic approaches with convincing efficacy for the treatment and prevention of DCVS and DCI after SAH. Despite various studies confirming the importance of inflammation in the pathophysiology of DCVS and DCI after SAH [1,3,4,23], the clear pathophysiology of the latter remains obscure. This is evident in the role of the different cytokines in the inflammation process and the related effects on neuronal cell death and on endothelial cells, which remain unclear. With this experimental study, we aimed to analyze the behavior of IL-1α in a rabbit closed cranium SAH model. The systemic and CSF IL-1α changes have never been described in a rabbit closed cranium SAH model.

In an ischemic stroke rat model, IL-1α, but not IL-1β, was expressed early on microglia-like cells in the ischemic hemisphere [24,26]. Moreover, IL-1α expression was closely associated with areas of focal blood–brain barrier breakdown and neuronal death, mostly near the penumbra surrounding the infarct, and therefore suggests that IL-1α is the major form of IL-1 contributing to inflammation early after cerebral ischemia [26]. In the context of SAH, a rat SAH model showed that haem induced the expression of IL-1α, produced by microglia and macrophages in the central nervous system, was present early after SAH throughout the brain [23,24,25,26,27,39]. Moreover pharmaceutically synthetized IL-1 Ra showed already promising anti-inflammatory effects in clinical trials of acute stroke and SAH [40,41]. Phase I and phase II clinical randomized trial studies showed that a reduction of IL-6 in serum and CSF and C-reactive protein in a SAH patient treated with an IL-1Ra, however, these studies were not sufficiently powered to analyze the clinical outcome of the patients [41,42]. Our results in this study showed that SAH was well simulated in all the rabbits in the SAH group with a significant increase of ICP and a decrease of CPP, and with a relevant presence of blood in the basal cistern and a significant presence of DCVS at follow up. These results are concordant with our previous studies, confirming the efficacy of the rabbit closed cranium SAH model simulating an aneurysmal SAH [1,30,31].

A correlation trend of CSF IL-1α, but not for serum IL-1α, with DCVS was found. Moreover, the level of systemic IL-1α were significantly higher at follow up in the SAH as compared with the sham group. The overall CSF IL-1α values were found to be higher than the serum values in both groups. This might be explained by a compartmental reaction in the CSF, with IL-1α release, possibly caused either by the surgical procedure itself or due to the generated SAH. Due to the compartmental inflammation reaction after SAH, CSF IL-1α might be more related to the development of DCVS than the serum IL-1α values (especially considering the higher levels of IL-1α in CSF than serum). The fact that serum IL-1α were higher in the SAH group at follow up, without showing a correlation with DCVS, might reflect the compartmental inflammation reaction happening in the CNS.

The results of this study were somewhat inconclusive, especially considering that we did not find any significant CSF increases of IL-1α in the SAH group between baseline and follow up, but only a correlation trend for DCVS. Those results might possibly be explained by the rather small sample size and the lack of power of the study, as this data were extrapolated from one larger study. We should also consider that IL-1α is a generic inflammation cytokine at the top of the inflammation cascade and that other cytokines, like IL-6 which triggers local inflammation and activation of the systemic acute phase response, might play an ever more specific role in the development of DCVS and DCI, as previously described [1,3,4]. Moreover, ELISA quantitative analysis of IL-1α were performed at day zero and day three after induction of SAH. Considering that IL-1α is an early expressed cytokine, possible measurement at day one and day two after SAH induction would have resulted in different values. Further experimental and clinical studies are surely warranted to better understand the behavior of IL-1α in the pathophysiology of DCVS and DCI and its relationship with neuronal cell death and interaction with other cytokines.

## 5. Conclusions

CSF IL-1α shows that a correlation trend with DCVS and systemic IL-1α is significantly elevated after SAH induction in a rabbit SAH model. IL-1α plays an important role at the beginning of the inflammation cascade in a rabbit closed cranium SAH model.

## Figures and Tables

**Figure 1 brainsci-09-00249-f001:**
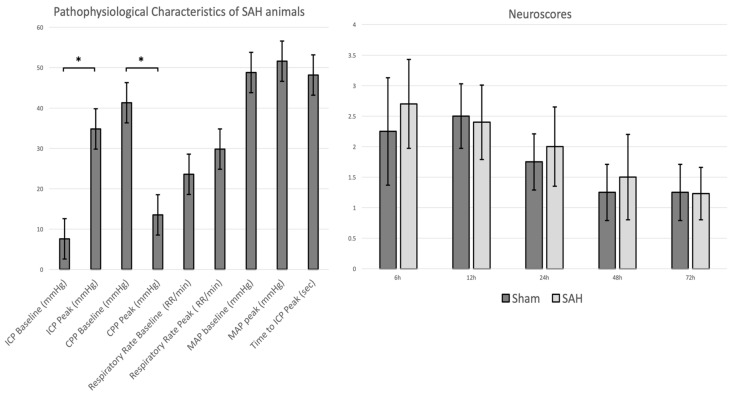
Pathophysiological characteristics of SAH animals (left panel) representing the changes of intracranial pressure (ICP), cerebral perfusion pressure (CPP), respiratory rate, and middle arterial pressure (MAP) at baseline (before SAH induction) and at peak (after SAH induction). Note, between baseline and peak there was a statistically significant increase of ICP and decrease of CPP, respectively. Data are presented as mean ± SD, *: *p* < 0.05. The right panel shows the neuroscores of the rabbits assessed at 6, 12, 24, 48, and 72 h after the surgical interventions. The SAH animals showed a tendency of worse neurological scores as compared to rabbits in the sham group (*p* = 0.27).

**Figure 2 brainsci-09-00249-f002:**
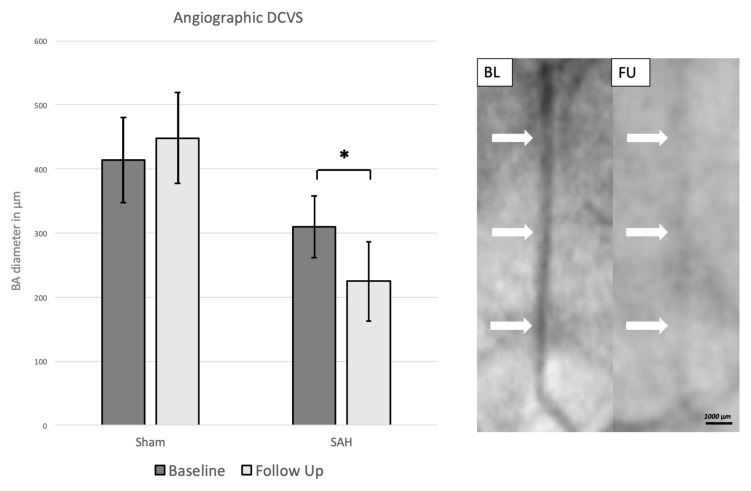
Left: Angiographic mean basilar artery diameter in µm at baseline and follow up. The mean basilar artery diameter decreased significantly at follow up in the SAH group. Data are presented as mean ± SD, *: *p* < 0.05. Right: DSA at baseline (BL) and follow up (FU) demonstrating decrease of caliber size at FU. White arrows: Point of measurement of basilar artery caliber showing decrease of size of the basilar artery at FU.

**Figure 3 brainsci-09-00249-f003:**
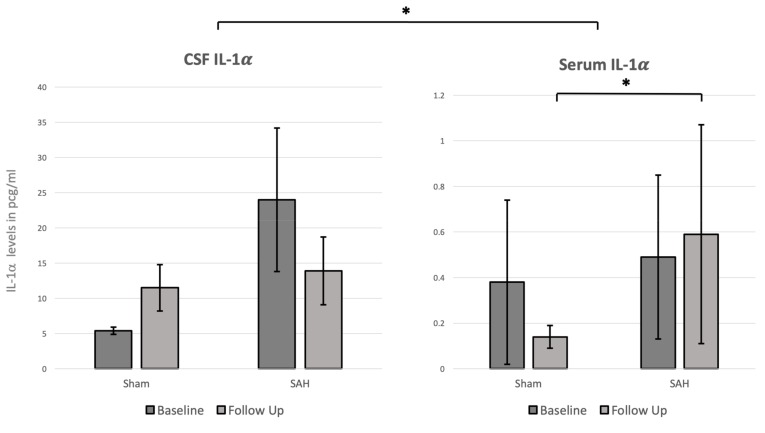
Mean baseline and follow-up IL-1α cerebrospinal fluid (CSF) and serum levels in pcg/mL. Note that the overall CSF IL-1α values were significantly higher as compared to the serum IL-1α levels. Moreover, the serum levels at follow up were significantly higher in the SAH group as compared to the sham group. Data are presented as mean ± SD, *: *p* < 0.05.

**Table 1 brainsci-09-00249-t001:** Baseline and follow-up analyses for the subarachnoid hemorrhage (SAH) and sham groups.

Parameters	SAH Group (*n* = 10)	Sham Group (*n* = 4)	*p*-Value
Baseline			
pH	7.3 (±0.1)	7.3 (±0.1)	0.310
pCO_2_ (mmHg)	77.1 (±12.5)	70.5 (±14.5)	0.215
pO_2_ (mmHg)	340.5 (±95.2)	308 (±150.6)	0.330
HCO_3_-mmol/L	28.9 (±7.1)	30.0 (±0.6)	0.325
BE mmol/L	10.8 (±5.9)	9.1 (±2.9)	0.301
SO_2_%	99.8 (±0.5)	99.5 (±0.7)	0.268
ctHb (g/dL)	12.3 (±5.3)	12.5 (±0.1)	0.323
Na^+^ mmol/L	143.3 (±2.2)	142 (±2.2)	0.164
K^+^ mmol/L	3.5 (±0.3)	3.47 (±2.2)	0.401
Ca^2+^ mmol/L	1.6 (±0.1)	1.5 (±0.1)	0.187
Glu mmol/L	15.8 (±3.4)	15.7 (±1.1)	0.487
Lac mmol/L	0.5 (±0.3)	0.5 (±0.1)	0.371
Heart rate/min	187.6 (±18.6)	175.7 (±3.8)	0.223
Middle Arterial Pressure	54.6 (±4.6)	63.7 (±0.3)	0.082
Weight Kg	3.73 (±0.3)	3.53 (±0.2)	0.183
**Follow up**	**SAH Group (*n* = 9)**	**Sham Group (*n* = 3)**	***p*-Value**
pH	7.3 (±0.1)	7.3 (±0.1)	0.349
pCO_2_ (mmHg)	70 (±14.3)	68.9 (±14.7)	0.457
pO_2_ (mmHg)	351 (±27.8)	431.3 (±29)	0.078
HCO_3_-mmol/L	13.2 (±6.2)	22.5 (±7.2)	0.138
BE mmol/L	7.3 (±7.2)	14 (±7.3)	0.186
SO_2_%	99.2 (±0.8)	99.5 (±0.7)	0.338
ctHb (g/dL)	10.8 (±0.8)	10.8 (±0.4)	0.50
Na^+^ mmol/L	142.7 (±3.8)	143 (±3.5)	0.435
K^+^ mmol/L	3.5 (±1.7)	3.9 (±1.5)	0.006
Ca^2+^ mmol/L	1.5 (±0.1)	1.5 (±0.1)	0.449
Glu mmol/L	18.8 (±4.1)	18.6 (±0.4)	0.428
Lac mmol/L	0.6 (±0.1)	0.6 (±0.2)	0.415
Heart rate/min	177.5 (±20.2)	184.5 (±7.5)	0.371
Middle Arterial Pressure	53 (±12.7)	57 (±12.7)	0.399
Weight Kg	3.52 (±0.4)	3.35 (±0.1)	0.07

Baseline and follow-up arterial blood gas analyses, middle arterial pressure, and weight in the SAH and sham groups (Legend: BE, base Excess; ctHb, concentration of hemoglobin; Glu, glucose, and Lac, lactate). Data are presented as mean ± SD.

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
