# Peer review of "Systemic and CSF Interleukin-1α Expression in a Rabbit Closed Cranium Subarachnoid Hemorrhage Model: An Exploratory Study"

_brainsci, 2019, doi:10.3390/brainsci9100249_

Round 1
Reviewer 1 Report
In the manuscript entitled “Systemic and CSF Interleukin-1α expression in a rabbit closed cranium subarachnoid haemorrhage model: An exploratory study” Croci and colleagues evaluated blood and cerebrospinal fluid (CSF) levels of IL-1α in a rabbit model of subarachnoid haemorrhage (SAH).
The authors found that the systemic levels of IL-1α were elevated after SAH induction. Although the authors did not find any significant CSF increases of IL-1α in the SAH group, the manuscript is somewhat interesting because supports the efficacy of this SAH model.
Some concerns need to be addressed.
In details:
The authors should explain the reason of the use of female animals. Mast cells have been associated to various neuroinflammatory conditions, including SAH [for a review see for example Parrella et al. Cells. 2019 May 10;8(5)]. Is there any link between IL-1 and mast cells? Please, discuss it briefly in the introduction. I suggest to include the mean bleeding sum score to figure 1. The quality of the figures appears poor. For example, in figures 1, 2 and 3 the writings are barely visible. Figures could be clearer by labelling the panels with letters. Finally I cannot see the scale bar in the figure 2, right panel. In Table 1 the spaces between the text appear too large. Line 96: “Detailed protocols for surgery, and angiography were previously described elsewhere12.” Twelve written as superscript looks like a citation. Please check.
Author Response
We thank reviewer#1 for his valuable comments and constructive criticism that has improved our manuscript. We have addressed the issues raised by the reviewer and modified the manuscript according to his helpful suggestions. All changes to the manuscript are marked in bold. We hope that the corrections made, together with the attached reply satisfy the concerns raised.
In our previous studies, using the SAH rabbit model, we used only female rabbits. To keep the model as standardized as possible we only used female rabbits for this study as well. Moreover, in the animal-research literature, the female sex is considered the stronger sex and it has been reported that female animals are less prone to diet-induced atherosclerotic lesions than male animals (Holm P, Circulation 1998), which is an important characteristic for our research question. Many thanks for this valuable comment. Indeed, mast cells have already been linked in different neuroinflammatory conditions and their presence has been described in the muscular layer of cerebral arteries after SAH (Pluta R.M er al. Res.2009). Regarding a potential link between IL-1 and mast cells: IL-1 stimulate the release of different cytokines and chemokines such as IL-6 from mast cells (Kandere-Grzybowska K. et al J. Immunol. 2003). As suggested, we added this important point in the introduction as follows: “IL-1 stimulates the release of IL-6 from mast cells, which have been described to be present in the aneurysm wall of SAH patients and muscular layer of cerebral arteries.” We adapted the quality of the images 1,2,3. Sadly we couldn’t add the sum score to figure 1 as it would make the quality of the image worse. As suggested we added a scale bar in figure 2. As suggested we adapted the table 1. We did verify the reference on line 96 and we did correct it.
Reviewer 2 Report
The authors undertake time consuming and otherwise well documented animal experiments to determine but 1 cytokine in csf and blood samples. The results do not quite support a major role for this cytokine in vasospasm pathogenesis after subarachnoid hemorrhage. It is mentioned that the study is part of a larger one. But it does not become clear what is investigated in the other parts of the study. It is not far to speculate that further cytokines or cytokine receptors are in the focus. Therefore, it would be appropriate to report on the entire data synoptically. Then known influences of cytikines and eventually new ones could be better estimated.
Author Response
We thank reviewer#2 for his valuable comments and constructive criticism that has improved our manuscript. Since this is an exploratory study to evaluate any possible changes of IL-1ain CSF and serum in a rabbit SAH model - with the goal to plan a larger study to investigate the interactions of different interleukins in the context of SAH - we cannot present further data at this point.
We used the rabbits from a larger study, analysing the inhibition of other inflammatory mediator, to respect the ethical principle of “reduction of the number of animals involved in experiments” based on the 3 “R” ethical principles rather than to design a study of its own (replacement, reduction, refinement) of animal experiments (Russel and Burch, The principles of Humane Experimental Technique, 1959).
Reviewer 3 Report
In this manuscript, the authors used a rabbit model to study systemic and CSF IL-1a level change after subarachnoid hemorrhage (SAH). In addition to the IL-1a expression, they also reported the arterial blood gases and basic physiological parameters, the intracranial pressure (ICP) and cerebral perfusion pressure (CPP) change before and after SAH induction, the neurological deficit scores, the basilar artery diameters. They concluded that the IL-1a level is elevated in the rabbit SAH model. The manuscript needs to be improved. My comments regarding this manuscript are as follows.
Major points
1) The significance of this study could be explained in more detail. The authors briefly mentioned the gap of knowledge in the abstract (line 21: The role of IL-1a has never been evaluated in a SAH-model) and the discussion (line 207: The systemic and CSF IL-1a change have never been described in a rabbit closed cranium SAH-model). Given that compiling evidence suggested IL-1 be a major therapeutic treatment for stroke, the contribution of the current study seems not very significant. But it could be improved if they also introduce the reason why it is important to close this knowledge gap under such background.
2) The method for evaluation of the neurological scores is missing. The method for correlation analysis is also missing. It would be better if the authors could put statistical analysis and ethic approval in different sections.
3) The neurological scores (Fig1 right panel) does not have SD, the difference between the sham and SAH group seems not statistically significant. Explain if this is expected and why.
4) The animal number are reduced in the follow up (SAH n=9, Sham n=3) compared to baseline (SAH n=10, Sham n=4). Provide an explanation for the reduction in numbers. In the abstract, the animal numbers are stated as SAH (n=12), Sham (n=5), they should be changed to the valid number of animals used in each group.
5) More animals are needed to draw a solid conclusion on IL-1a level change in the CSF. It seems that the IL-1a level in SAH group is profoundly higher at baseline, and not much different from the sham group at follow up, the authors report a trend of increase with no statistical significance. Since the IL-1a level is the major finding in the current research, a trend of change is not enough to support the conclusion.
Minor points
1) The figs need to be improved. For example, Fig2 and Fig 3 do not have a y-axis title and unit. Fig 2 right panel, baseline was abbreviated as “BA” in the legend but “BL” in the graph. The legend in the figs (Sham & SAH) are very small and hard to read when they are in the bottom.
2) In the statistical analysis section, it stated in line 127 “Continuous values were given as mean±SEM”, however in figs, all numbers are shown as mean±SD.
3) Revise the whole manuscript for typos and mistakes. For example: line 31 “IL-1a level are elevated”. Line 55 “*** IL-1bin ***”. Revise the long sentence (line54-56) “IL-1a differs mainly ******both on the cell surface and intracellularly [18].” Revise sentence (line 171-172) “The sham group showed *****baseline respectively follow up (p=0.446) respectively (Figure 2)”
Author Response
Answer to reviewer #3:
We thank reviewer# 3 for his valuable comments and constructive criticism that has improved our manuscript. We addressed the issues raised by the reviewer and modified the manuscript according to his helpful suggestions. All changes to the manuscript are marked in bold. We hope that the corrections made, together with the attached reply satisfy the concerns raised.
Major points:
1) We corrected the sentence in the abstract “the role of IL-1αhas never been evaluated in a SAH-model” with “the role of IL-1αhas never been evaluated in a rabbit-SAH-model”.
IL-1 is one of the earliest interleukin released in context of sterile inflammation. So far, no medical treatment has been established to prevent DCI and poor outcome after SAH (MacDonald RL et al., Stroke 2012, Muroi et al. Curr Opin Crit Care 2012). Inflammation is considered a key player in early brain injury after SAH and there is an urgent need of better understanding the involved pathophysiological inflammation cascade (Marbacher et al. Transl. Stroke Res 2014, Muroi et al. Neurosurgery 2013). IL-1αdemonstrated to be early expressed in areas of focal neuronal injury after ischemic injuryin animal stroke models (Luheshi, N, Journal of Neuroinflammation 2011). Therefore, if it is confirmed in SAH model as well, it could represent a novel target for possible therapeutic approaches to prevent DCVS and DCI after SAH. We introduced the specific reason of why it is important to close this knowledge gap of IL-1α in the pathophysiology of DCVS and DCI after SAH in the Introduction section (Line 94-101, Page 4).
2) As suggested, we added the method for evaluation of the neurological scores and the method for correlation analysis (Line 159, page 6). We put statistical analysis and ethic approval in two different sections: 2.4 Statistical analysis and 2.5 Ethic approval (Line 165, page 7).
3) As suggested, we added the SD to the neurological score (Figure 1, right panel). We added the p value to the text (line 188, page 7)and to the legend of Figure 1. Like in our previous studies using this model we didn’t expect any large difference in neurological scores between the two groups. The score used, analyses presence of severe focal neurological deficits. This is a rare phenomenon despite presence of severe vasospasm in rabbits. This is mainly due to abundant collaterals of the rabbit brain. Despite this limitation the applied neuro-score provides a rough overview of them animals clinical behaviour after SAH.
4) We started with a total of 17 NZW Rabbits. One rabbit was prematurely excluded after he died before the start of the group-specific procedure. In two rabbits, premature euthanasia was performed due to a severe postoperative neurological deficit after SAH induction, therefore 10 rabbits in the SAH and 4 in the sham group remained for arterial blood gas analysis at baseline. At follow up because of a technical issue with the arterial blood gas analysis we didn’t have values for a rabbit in the SAH and one in sham and in the group, therefore a n=9 in the SAH group and n=3 in the sham group at follow up for the blood gas analysis. The drop outs are given in detail in the Results Section on First Paragraph (Line 186, Page 7).
5) We agree that the n of this study is low to draw definitive conclusion about the changes of IL-1ain this rabbit SAH model. We agree as well that the IL-1aCSF values inthe SAH group are profoundly higher at baseline, and not much different from the sham group at follow up, in fact we have already state it in the paper: “in the SAH group there was a trend of higher CSF IL-1alevels at baseline and follow up compared to the sham group without statistical significance” (Line 202, Page 8). Or conclusion was based on the systemic IL-1avalues and not IL-1aCSF values. Whencomparing the systemic IL-1avalues in the serum atfollow up there was a significant increase of systemicIL-1ain the SAH compared to the Sham group (p = 0.042)
Minor Points:
1) We added the y-axis title to figure 2 and 3. We corrected BA with BL in the legend of Figure 2. We enlarged the legends in figure “sham and SAH”.
2) We apologize for the mistake in the statistical analysis section, “Continuous values were given as mean ± SEM”, we did correct it with “mean ± SD”
3)We revised the manuscript and corrected the different typos and mistakes.
Round 2
Reviewer 3 Report
I would like to thank the authors for addressing my initial comments. However, two responses need additional details and I have additional comments regarding the responses.
1) Neurological scores:
“As suggested, we added the SD to the neurological score (Figure 1, right panel). We added the p value to the text (line 188, page 7)and to the legend of Figure 1. Like in our previous studies using this model we didn’t expect any large difference in neurological scores between the two groups. The score used, analyses presence of severe focal neurological deficits. This is a rare phenomenon despite presence of severe vasospasm in rabbits. This is mainly due to abundant collaterals of the rabbit brain. Despite this limitation the applied neuro-score provides a rough overview of them animals clinical behaviour after SAH.”
As the authors has state that they do not expect large difference in neurological scores between the two groups. Are we expecting a difference or not? Are there studies from other research groups that support your statement? If so, these studies needs to be included to support your result in order to be clear what is expected.
2) CSF and systemic IL-1al evels:
“ We agree that the n of this study is low to draw definitive conclusion about the changes of IL-1ain this rabbit SAH model. We agree as well that the IL-1aCSF values inthe SAH group are profoundly higher at baseline, and not much different from the sham group at follow up, in fact we have already state it in the paper: “in the SAH group there was a trend of higher CSF IL-1alevels at baseline and follow up compared to the sham group without statistical significance” (Line 202, Page 8). Or conclusion was based on the systemic IL-1avalues and not IL-1aCSF values. Whencomparing the systemic IL-1avalues in the serum atfollow up there was a significant increase of systemicIL-1ain the SAH compared to the Sham group (p = 0.042)”
Serum IL-1a and its role in regulating DCVS was not properly explained in the introduction or discussion. This is an abrupt change in conclusion. Is there a correlation between serum IL-1a and DCVS which the authors initially wanted to study? The pattern of change in CSF IL-1a level seem very different from the serum IL-1a level. Is it the serum IL-1aor CSF IL-1a that is implicated in DCVS? What is the conclusion regarding the initial aim of the study stated in introduction? It is important to answer those questions clearly in the text and draw a clear and well-interpreted conclusion.
Author Response
Answer to reviewer #3 (Round 2):
We thank reviewer#3 for his valuable additional comments and constructive criticism that further improved our manuscript. We have addressed the issues raised by the reviewer and modified the manuscript according to his helpful suggestions. All changes to the manuscript are marked in bold. We hope that the corrections made, together with the attached reply satisfy the concerns raised.
According to results of our previous studies (Andereggen et al. 2014; Marbacher et al. 2014; Andereggen et al. 2014; Andereggen et al. 2014; Marbacher et al. 2015; Croci et al. 2016) we didn’t expect any significant difference of the neurological scores between the groups. To our knowledge there aren’t other studies which used the exact same rabbit subarachnoid haemorrhage model and the same neurological score. As stated in the introduction we aimed to evaluate the levels of systemic (serum) and CSF IL-1a in an established extracranial intracranial closed rabbit SAH model and to analyse if there is any correlation with DCVS. CSF IL-1a levels were significantly higher than serum IL-1a. Regarding if is the serum IL-1a or CSF IL-1a that is implicated in DCVS: we found a correlation trend between CSF IL-1a and DCVS and not between serum IL-1a and DCVS. Therefore, CSF IL-1a might be more implicated in the development of DCVS. The higher levels of serum IL-1a in the SAH group might reflect the compartment inflammation reaction initiated by blood in the subarachnoid space. We adapted the paper regarding this issue in the “discussions” (Line 243-247, Page 8) and in the “conclusions” (Line 262-264, Page 8).